# Anti-Inflammatory Actions of G-Protein-Coupled Estrogen Receptor 1 (GPER) and Brain-Derived Estrogen Following Cerebral Ischemia in Ovariectomized Rats

**DOI:** 10.3390/biology12010099

**Published:** 2023-01-09

**Authors:** Jing Xu, Jing Bai, Fujia Gao, Chao Xu, Yuanyuan Huang, Danyang Li, Lu Wang, Ruimin Wang

**Affiliations:** 1Dementia and Dyscognitive Key Laboratory, Tangshan 063210, China; 2School of Basic Medical Sciences, Hebei Key Laboratory for Chronic Diseases, North China University of Science and Technology, Tangshan 063210, China; 3Neurobiology Institute, School of Public Health, North China University of Science and Technology, Tangshan 063210, China

**Keywords:** global cerebral ischemia, G-protein-coupled estrogen receptor 1, aromatase, brain-derived estrogen, inflammation

## Abstract

**Simple Summary:**

The overall goal of this study was to examine the role of G-protein-coupled estrogen receptor 1 signaling in the regulation of the early inflammatory process following global cerebral ischemia. The results of the study reveal that G-protein-coupled estrogen receptor 1 signaling reduces pro-inflammatory signals and enhances anti-inflammatory signals in the brain at early timepoints after global cerebral ischemia. Furthermore, the study shows that brain-derived estrogen helps to mediate the anti-inflammatory effects of G-protein-coupled estrogen receptor 1 signaling. As a whole, the study demonstrates an important role and synergy of G-protein-coupled estrogen receptor 1 signaling and brain-derived estrogen in the control of early inflammatory events after global cerebral ischemia. An increased understanding of the early-stage regulation of inflammation after global cerebral ischemia could lead to new and improved therapies to protect the brain from ischemic damage and improve outcomes after global cerebral ischemia.

**Abstract:**

Global cerebral ischemia can elicit rapid innate neuroprotective mechanisms that protect against delayed neuronal death. Brain-derived 17β-estradiol (BDE2), an endogenous neuroprotectant, is synthesized from testosterone by the enzyme aromatase (Aro) and is upregulated by brain ischemia and inflammation. Our recent study revealed that G1, a specific G-protein-coupled estrogen receptor 1 (GPER) agonist, exerts anti-inflammatory and anti-apoptotic roles after global cerebral ischemia (GCI). Herein, we aimed to elucidate whether G1 modulates the early inflammatory process and the potential underlying mechanisms in the ovariectomized rat hippocampal CA1 region. G1 was found to markedly reduce pro-inflammatory (iNOS, MHCII, and CD68) and to enhance anti-inflammatory (CD206, Arginase 1, IL1RA, PPARγ, and BDNF) markers after 1 and 3 days of reperfusion after GCI. Intriguingly, the neuroprotection of G1 was blocked by the Aro inhibitor, letrozole. Conversely, the GPER antagonist, G36, inhibited Aro-BDE2 signaling and exacerbated neuronal damage. As a whole, this work demonstrates a novel anti-inflammatory role of GPER, involving a synergistic mediation with BDE2 during the early stage of GCI.

## 1. Introduction

Global cerebral ischemia (GCI), which can occur due to cardiac arrest, leads to delayed cell death of neurons in the hippocampal CA1 region in both experimental animals and patients [1,2]. Inflammation is one of the most noted and crucial pathogenesis of ischemic brain injury. Macrophages, which are derived from peripheral blood, and activated microglia localized in the brain are the primary participants in inflammation [3,4,5]. There is increasing evidence that the brain microenvironment after ischemic injury can induce the polarization of microglia into two different states: the classic activated M1-like phenotype and the alternative M2-like phenotype [6]. The M1 phenotype of microglia can produce pro-inflammatory factors such as inducible nitric oxide synthase (iNOS), cluster of differentiation 68 (CD68), interleukin-1 beta (IL1β), interleukin 6 (IL6), and tumor necrosis factor-alpha (TNFα), which impair neurons and glial cells and worsen inflammatory injury. In comparison, the M2 phenotype can release anti-inflammatory and pro-survival factors, such as Arginase 1, cluster of differentiation 206 (CD206), interleukin 10 (IL10), IL1 receptor antagonist (IL1RA), transforming growth factor-beta (TGF-β), brain-derived neurotrophic factor (BDNF), and glia-derived neurotrophic factor (GDNF), to attenuate inflammation and promote brain repair after ischemic insult [7,8]. Therefore, the degree of M1/M2 imbalance is linked to the severity of brain injury, suggesting that the alteration in the balance of the ratio might serve as a potential therapeutic strategy for ischemic brain injury.

17β-Estradiol (E2), as a powerful neuroprotectant, is involved in attenuating the inflammatory process, while the effect of E2 microglia/microphage polarization remains controversial [6,9,10]. Our recent study revealed that exogenous low-dose E2 can directly suppress inflammation via the mediation of microglial activation and M1–M2 phenotype switch after 7 days of reperfusion after GCI [6]. However, there is evidence to suggest that the chronic administration of E2 to ovariectomized (OVX) mice can significantly increase M1 phenotype markers such as IL1β, IL6, TNFα, and iNOS [9,10]. While E2 is generally thought of as a primarily ovarian-generated hormone that circulates in the blood to regulate target tissues throughout the body, there is growing evidence that E2 is also a neurosteroid produced by astrocytes and neurons in the brain and can act as an important endogenous neuroprotectant [11]. Aromatase (Aro), the key enzyme in the synthesis of brain-derived E2 (BDE2), is mainly expressed in resting neurons; however, its expression can be markedly upregulated in astrocytes after brain injury or ischemia [12,13]. Subsequent studies revealed that E2, the selective estrogen receptor mediator (SERM) tamoxifen, and dietary supplementation with docosahexaenoic acid or genistein can induce Aro expression [11,14,15,16]. Furthermore, various studies, including our own, confirmed the neuroprotective actions of Aro-BDE2 via the attenuation of neuroinflammatory and excitotoxic damage [13,17,18,19], as well as the mediation of synaptic plasticity [20,21]. However, the precise neuroprotective roles and the potential mechanisms of Aro-BDE2 signaling following GCI have not been fully elucidated.

It is well known that E2 plays a neuroprotective role by binding to estrogen receptors (ERs), including ERα and -β, and G-protein-coupled estrogen receptor 1 (GPER/GPR30). Dillion et al. investigated tissue- and sex-specific differences in ER mRNA, revealing that GPER mRNA is relatively stable in both sexes and predominantly present in the key brain regions of the hippocampus, somatosensory cortex, and prefrontal cortex, where lower expression levels of ERα/β exist [22]. Therefore, our group has recently focused on elucidating the anti-inflammatory effects and potential mechanisms induced by GPER following GCI. G1, a specific agonist of GPER, serves as an alternative neuroprotectant for E2 and exerts neuroprotective effects by inhibiting microglial activation mediated by Toll-like receptor 4 (TLR4) [23]. Recently, our group showed that GPER activation induced by G1 reduces the inflammatory damage of neurons in the hippocampal CA1 region after GCI by reducing NLRP3 inflammasome activity and upregulating IL1RA. The neuroprotective effect of G1 following GCI appears to be mediated by GPER, as it is blocked by the GPER antagonist, G36 [24].

Since GPER is expressed in neurons, astrocytes, and microglia in the brain [24,25,26], the current study examined whether GPER may have a role in mediating the anti-inflammatory and neuroprotective effects of BDE2 in the hippocampus following cerebral ischemia. Furthermore, since virtually nothing is known about the GPER regulation of the early inflammatory process that occurs 1–3 days after GCI, we focused our study on this critical time period, which is thought to play an important role in delayed neuronal cell death and pathology after GCI.

## 2. Materials and Methods

### 2.1. Animals and Animal Model of Global Cerebral Ischemia (GCI)

Adult female ovariectomized Sprague-Dawley rats (Beijing HFK Biotechnology Co., Ltd., Beijing HFK Biotechnology Co., Ltd., Changping, Beijing, China; 3 months old) were maintained in a temperature-controlled room (22–24 °C), fed freely, and kept in accordance with a 12 h/12 h light cycle. We used ovariectomized rats to create an estrogen-deficient environment to study the effects of the GPER agonist, G1, without the confound of high circulating estrogen levels, which are present in intact female rats. From a translational viewpoint, the ovariectomized rat most resembles the surgical menopausal human whose the ovaries are removed early due to medical reasons. After bilateral ovariectomy (OVX) under isoflurane anesthesia, all rats were divided randomly into 5 groups: sham operation group (sham), ischemia-reperfusion group (I/R), GPER specific agonist G1 treatment group (I/R + G1), GPER specific antagonist G36 treatment group (I/R+G36), and G1 plus aromatase inhibitor letrozole treatment group (I/R + G1 + Let). GCI was induced after one week of OVX as in our previously described study [24]. Briefly, the bilateral vertebral arteries of the rats were electrocauterized, and 24 h later, the bilateral common carotid arteries were clipped with artery clamps for 12 min. The rats that showed pupil dilation and lack of response to light during ischemia, as well as loss of their righting reflex within 30 s, were considered as successful and used for the experiments. The resumption of carotid artery blood flow was visually verified upon releasing the clips. The body temperature of the animals was maintained at 36.5–37.5 °C with an electric blanket during the experiment. Sham-operated animals were handled in the exact same was as ischemic animals, except that they did not have their common carotid arteries clamped.

### 2.2. Administration of Drugs

GPER agonist G1 (Tocris, Bristol, UK; Cat. No. 3577; 10 μg/day) or GPER antagonist G36 (Tocris; Cat. No. 4759; 10 μg/day) was administrated subcutaneously using a minipump (0.5 µL/h, 14 days release; Alzet Model 2002) that was implanted at the beginning of the OVX operation. A vehicle treatment group that received the same volume of cottonseed oil with 1% DMSO, which was administrated at the same time as G1 or G36 administration, was included in the study. In the letrozole treatment group, the rats received, once daily, intragastric administration of letrozole (Sigma-Aldrich, St. Louis, MI, USA; PHR1540) at a concentration of 10 mg/kg dissolved in 1% carboxymethlycellulose.

### 2.3. Hippocampal Tissue Preparation and Immunofluorescence Staining

To anesthetize the rats, isoflurane was administered after 1 d or 3 d of reperfusion. Pericardial transfusion was performed (0.9% saline with 4% paraformaldehyde in 0.1 M phosphate buffer (PB; pH 7.4)). The post-fixing of the brains was performed overnight in the same fixative at 4 °C, followed by gradient dehydration in 10%, 20%, and 30% sucrose in 0.1 M PB. Coronal sections (25 μm) were then cut with a cryostat. Continuous sections were collected through the entire dorsal hippocampus (2.5–4.5 mm posterior to bregma). The brain sections were used for immunofluorescence staining as in our previous work [24]. In brief, the sections were washed for 30 min in 0.1 M PB, permeabilized for 1 h in 0.4% Triton X-100-PBS, blocked for 2 h in 10% donkey serum, and then incubated with the following primary antibodies at 4 °C overnight: iNOS (ARIGO; 56509; 1:80), Iba1 (ab5076; 1:1000), MCHII (ab180779; 1:250), CD68 (ab31630; 1:500), CD206 (ab64693; 1:500), Arginase 1 (sc166920; 1:50), IL1RA (ab124962; 1:500), PPARγ (NBPI-61399; 1:100), BDNF (ARIGO 56653; 1:200), ARO-CYP19 (sc30086; 1:100), Neuronal nuclei (NeuN) (Millipore, Burlington, MA, USA; NG1857584; 1:300), and Estradiol (BioGenex, Fremont, US; ARO380913). After incubation with the corresponding primary antibody, the sections were washed using 0.1% TritionX-100-PBS for 40 min at room temperature and then incubated with secondary antibodies (Alexa-Fluor 488/568/647 donkey anti-rabbit/anti-mouse/anti-goat (1:250; Invitrogen, Waltham, MA, USA)) at room temperature for 1.5 h. After washing in 0.1% TritionX-100-PBS for 50 min, the sections were mounted on slides and counterstained with DAPI (Lot ZA0210; Vector Laboratories, Inc., Burlingame, CA, USA, 94010). A laser scanning confocal microscope was used to capture confocal images (LSCM, Olympus Corporation, Tokyo, Japan; Olympus FV1000) and digital imaging software (FV10-ASW 1.5 Viewer).

### 2.4. Hippocampal Sample Preparation and Western Blot Analysis

The rats were sacrificed under deep anesthesia with isoflurane 1 d and 3 d after ischemia, and the brains were rapidly removed. The hippocampal CA1 regions were micro-dissected on an ice pad and stored at −80 °C until used. The hippocampal CA1 tissues were homogenized with a high-efficiency Radio Immunoprecipitation Assay (RIPA) lysate (Solarbio, Beijing, China; R0010) on ice using a Teflon–glass homogenizer. Protease and phosphatase inhibitors (Thermo Scientific, Rockford, IL 150825, USA) were added to the lysate before homogenization. The homogenates were centrifuged at 15,000× *g* for 30 min at 4 °C to obtain a total fraction in the supernatants. An enhanced BCA Protein Assay Kit (Multi Sciences, Hangzhou, China) was used to determine protein concentrations, where bovine serum albumin (BSA) was used as the standard.

Protein samples were heated for 5 min (at 100 °C) with loading buffer containing 0.125 M Tris-HCl (pH 6.8), 20% glycerol, 4% SDS, 10% mercaptoethanol, and 0.002% bromophenol blue. A total of 30 μg protein per line was separated using sodium dodecyl sulfate-polyacrylamide gel electrophoresis (10–12% SDS-PAGE) and transferred to a PVDF membrane. Blocking was then performed for 1 h in 3% BSA, followed by overnight incubation of the PVDF membrane at 4 °C with the following primary antibodies: iNOS (ARIGO; 56509; 1:800), MCHII (ab180779; 1:500), CD68 (ab31630; 1:500), CD206 (ab64693; 1:500), Arginase 1 (sc166920; 1:200), IL1RA (ab124962; 1:1000), PPARγ (NBPI-61399; 1:200), BDNF (ARIGO; 56653; 1:200), ARO-CYP19 (sc30086; 1:500), IL4 (Gentex; 155806; 1:5000), GAPDH (sc32233; 1:1000), and Tubulin (sc9104; 1:200). Membrane washing was then performed for at least 30 min using 0.2% Tween-20 in Tris-buffered saline (TBST). The membranes were then incubated for 1.5 h at room temperature with HRP-conjugated secondary antibodies. A CCD digital imaging system was used to visualize bound proteins. Image J 1.49 analysis software was used to perform semi-quantitative analyses of the bands. Next, the normalization of band densities of the targeted proteins to the loading controls (GADPH or β-tubulin) was performed. Means ± SEs were calculated from the data.

### 2.5. Aromatase Activity

Aromatase activity from the protein samples from sham, I/R1d, I/R1d + G36, I/R1d + G1, and I/R1d + G1 + Let rats were detected using an assay kit (fluorometric) (ab273306) according to the protocol provided by the manufacturer. In brief, after preparing the reagents (samples, aromatase substrate, aromatase inhibitor Letrozole, positive control, fluorescence standard, et al.), aromatase reaction mixes (2×, 50 µL) were prepared by combining 30 µL of sample (50 µg of protein per well) and 2 µL of NADPH generating system in a 96-well plate and adjusting the final volume to 50 µL with aromatase assay buffer. Additionally, background control (no enzyme), positive inhibition control (sample + letrozole), positive control (PC; recombinant human aromatase, 25 µL), and PC + inhibitor Letrozole (5 µM) were needed. Then, the final volume of each well was adjusted to 70 µL with aromatase assay buffer. Following incubation for 10 min at 37 °C, an aromatase substrate/NAP+ mixture (30 µL) was added to each well using a multichannel pipette, yielding a final reaction volume of 100 µL per well. The plate was incubated at 37 °C for 60 min; then, fluorescence was immediately measured in kinetic mode at EX/EM = 488/527 nm using SpectraMax M5 Multi-Mode Microplate Reader (Molecular Devices, San Jose, CA, USA). A highly selective aromatase inhibitor (>100-fold selectivity for aromatase over other enzymes) was used to determine aromatase activity in heterogeneous biological samples where other CYP isozymes may contribute to substrate metabolism. The calculation of aromatase-specific activity was performed by running parallel with and without the inhibitor and subtracting any residual activity detected with the inhibitor present.

### 2.6. Statistical Analysis

For the statistical analysis of the data, either one-way or two-way analysis of variance (ANOVA) was performed, followed by the Student–Newman–Keuls method using SigmaStat 3.5 software. Results are shown as means ± SEs. GraphPad Prism 8 software was used to generate the statistical figures. Differences were considered significant at #, ##, and ###, denoting *p* < 0.05, *p* < 0.01, and *p* < 0.001, respectively.

## 3. Results

### 3.1. G1 Administration Suppresses M1 Phenotypic Markers of Microglia in the Hippocampal CA1 Region Following GCI

Ionized calcium binding adaptor molecule 1 (Iba1) is a cytoskeleton protein specific to microglia and macrophages. Our previous study showed that Iba1 immunoreactivity was significantly enhanced after 1 d and 3 d of reperfusion after GCI [6]. Therefore, we selected 1 and 3 days as timepoints to examine the effects of G1 administration on the expression of phenotype markers for the M1 pro-inflammatory type of microglia in OVX rats. The Western blot analysis showed that the protein expression levels of M1 phenotype markers iNOS (Figure 1A(a1,a2) and Appendix A), MHCII (Figure 1B(b1,b2) and Appendix A), and CD68 (Figure 1C(c1,c2) and Appendix A) were significantly increased after 1d and 3d of reperfusion (I/R1d and I/R3d) compared with sham animals, and G1 administration markedly suppressed the expression of M1 markers in both the I/R1d and I/R3d groups. The results of the immunofluorescent (IF) staining of M1 markers with Iba1 are shown in Figure 1D–F. M1 marker staining is indicated by the green color, while IBA1 staining is indicated by the red color. As shown in Figure 1D–F, Iba1-positive cells in the hippocampal CA1 region had a large and round body in I/R1d and I/R3d animals, which is consistent with the typical ameboid shape that microglia exhibit upon activation. In contrast, Iba1-positive cells in sham and G1-administered animals exhibited a ramified, branched appearance, typical of resting microglia. In addition, the immunofluorescence intensity of iNOS, MHCII, CD68, or Iba1 was markedly enhanced in I/R1d and I/R3d groups compared with the G1-treated groups at the same timepoint. Meanwhile, G1 administration significantly reduced the co-localization of Iba1 with iNOS (yellow, Figure 1D) and of Iba1 with CD68 (yellow, Figure 1F) compared with the I/R group at the same timepoint. These results indicate that G1 administration can suppress early-phase inflammation, potentially by inhibiting microglial activation after GCI.

### 3.2. G1 Administration Enhances M2 Markers of Microglia in the Hippocampal CA1 Region Following GCI

Next, the effect of G1 administration on the protein expression of anti-inflammatory M2 phenotype markers CD206, Arginase 1, and IL1RA was examined using Western blot and IF staining analyses. The study demonstrated that the protein expression levels of two of the three M2 phenotype markers (CD206 and IL1RA) were significantly decreased in I/R1d and I/R3d animals compared with sham animals, while G1 administration strongly elevated the protein expression of all M2 markers (Figure 2A–C(a1–c2) and Appendix A). Arginase 1 protein expression was not significant changed after 1d of reperfusion; however, it markedly decreased after 3d of reperfusion compared with sham animals, and G1 elevated Arginase 1 levels compared with the I/R group (Figure 2B(b1,b2) and Appendix A). We next conducted double IF staining of M2 markers with Iba1, and the results showed that G1-treated animals had higher IF intensity of M2 markers than the I/R groups at the same timepoints (Figure 2D–F). These results indicate that G1 can elevate the M2 anti-inflammatory microglial phenotype in the hippocampal CA1 region during the early phase of GCI.

### 3.3. G1 Administration Increases the Levels of Brain-Derived Neurotrophic Factor (BDNF) and Pro-Survival Transcription Factor PPARγ in the Hippocampal CA1 Region after GCI

Brain peroxisome proliferator-activated receptor gamma (PPARγ) is classified as part of the nuclear receptor superfamily of ligand-dependent transcription factors. PPARγ can enhance the expression of the neuroprotective factor, BDNF, and has been implicated in neuroprotection against inflammatory impairment [27,28,29]. We thus next examined the expression of PPARγ and BDNF using Western blot and IF analyses. As shown in Figure 3A,B, a significant decrease in the protein expression of PPARγ and BDNF was demonstrated in I/R1d compared with the sham group, while G1 administration markedly enhanced the protein expression of both PPARγ and BDNF compared with I/R groups at the same timepoint (Figure 3A,B(a1–b2) and Appendix A). Representative photomicrographs of IF staining of PPARγ (red) with Iba1 (green) or BDNF (red) are shown in Figure 3C,D. The IF staining results closely mirrored the findings of the Western blot analysis, as G1 elevated the immunofluorescent intensity of both PPARγ and BDNF in I/R1d and I/R3d.

### 3.4. GPER Plays an Important Role in the Neuroprotective Effect of Aromatase-BDE2 Signaling on the Hippocampal CA1 Region after GCI

Recent work has demonstrated that aromatase protein and BDE2 levels are upregulated in response to brain insults such as GCI and brain trauma and that BDE2 exerts anti-inflammatory/neuroprotective effects [11,19,21]. Since GPER is expressed in neurons and glia, we examined whether GPER may mediate the increase in Aro and BDE2 after GCI. The strategy we chose to use for this study was to block GPER activation using the GPER antagonist, G36, and examining the effect on Aro protein and its activity following GCI. As shown in Figure 4A and Appendix A, the I/R1d group exhibited significantly increased protein expression of Aro compared with sham animals, while G36 markedly reversed this enhancement in the hippocampal CA1 region. Additionally, we performed double IF staining of Aro (green) and E2 (red) and found that the fluorescence intensities of Aro and E2 were significantly increased in the hippocampal CA1 region in the I/R1d group versus the sham group, while G36 blocked this effect (Figure 4B(b1,b2)). Furthermore, Aro activity in the hippocampal CA1 region was detected using an enzyme-linked immunosorbent assay (ELISA). The results closely matched those of the Western blot analysis for Aro protein, as they showed an enhancement in Aro activity in the I/R1d group compared with the sham group, and G36 significantly reversed this effect (Figure 4C). These results indicate that GPER signaling is necessary for increase in Aro protein and its activity in the hippocampus after GCI.

Since BDE2 promotes anti-inflammatory and neuroprotective effects after GCI, we next examined whether GPER signaling could mediate these beneficial effects of BDE2. To accomplish this goal, we examined microglial activation using IF staining of Iba1, while neuronal survival was assessed using IF staining of the neuron marker, NeuN. As shown in Figure 4D, IF staining revealed that there were no significant differences in the number of NeuN-positive cells in the hippocampal CA1 region between the sham and I/R1d groups, although the immunofluorescence intensity of Iba1 was significantly increased in the I/R1d group compared with the sham group. Intriguingly, G36 treatment not only robustly elevated Iba1 immunoreactivity and caused Iba1-positive cells to display an ameboid morphology, but also significantly decreased the number of NeuN-positive cells (Figure 4D(d1,d2)). These results suggest that GPER signaling helps to mediate the anti-inflammatory and neuroprotective actions of BDE2 in the hippocampus after GCI.

### 3.5. The Anti-Inflammatory Role of G1 Is Dependent upon Aro-BDE2 Signaling in the Hippocampal CA1 Region Following GCI

Next, we used the Aro inhibitor, letrozole (Let), to explore whether Aro-BDE2 signaling is required for the anti-inflammatory effect of G1 in OVX rats. Western blot analysis (Figure 5A and Appendix A) revealed no significant differences in the protein expression of Aro among the three groups (I/R1d, G1, and G1+Let) in the hippocampal CA1 region. Furthermore, triple IF staining of Aro (green), Iba1 (blue), and NeuN (red) confirmed the Western blot results, showing no differences in Aro fluorescence intensity among the three groups (Figure 5B(b1)). Notably, Aro was mainly expressed in glia-like cells in I/R1d and Let-treated animals, while it was specifically expressed in NeuN-positive cells in G1-treated animals (Figure 5B). Intriguingly, G1 significantly attenuated the IF intensity of Iba1 compared with I/R1d animals, and Let treatment blocked the effect of G1 (Figure 5B(b2)). Correspondingly, the number of NeuN-positive cells in the hippocampal CA1 region of Let-treated animals was significantly decreased, while there were no significant differences between the I/R1d and G1 treatment groups (Figure 5B(b3)). Finally, Aro activity was examined with an ELISA analysis. As shown in Figure 5C, Let treatment markedly attenuated Aro activity, while no significant changes were observed between the I/R1d and G1-treated groups. These results indicate that Aro and BDE2 may contribute to the anti-inflammatory effect of G1.

### 3.6. Synergistic Action of G1 and BDE2 Promotes Protein Expression of IL-4, PPARγ, and BDNF in the Hippocampal CA1 Region in the Early Phase of GCI

We next used the GPER-specific antagonist, G36, to further examine the role of GPER in regulating the expression of anti-inflammatory factors interleukin 4 (IL4), PPARγ, and BDNF. As shown in Figure 6A(a1–a3) and Appendix A, G36 significantly decreased the protein expression levels of anti-inflammatory factors IL4 and PPARγ, and neurotrophic factor BDNF compared with the I/R1d group. To determine whether the G1 effects on the expression of IL4, PPARγ, and BDNF involve BDE2, we utilized the aromatase inhibitor, letrozole (Let). The results revealed that Let treatment markedly reversed the effects caused by G1 treatment, producing significant attenuation of the protein expression levels of IL4, PPARγ, and BDNF (Figure 6B(b1–b3) and Appendix A). Overall, the results reveal a synergistic action of G1 and BDE2 to elevate the protein levels of anti-inflammatory factors IL4, PPARγ, and BDNF in the hippocampal CA1 region.

## 4. Discussion

In the current study, we used ovariectomized rats to study the effect of GPER activation. Ovariectomized rats were chosen because a previous study by our group revealed that GPER activation in ovariectomized rats can protect hippocampal pyramidal neurons from ischemic insult by reducing NLRP3 inflammasome activity in microglia and enhancing anti-inflammatory factor IL1RA in hippocampal neurons 14 days after GCI [28]; we thus wished to explore changes at earlier timepoints that might explain the neuroprotective effects of G1. It should be noted that studies in ovariectomized animal models are much needed and important, as it is well known that surgical menopausal patients, which have had their ovaries removed prior to menopause, are at significantly increased risk of ischemic stroke, double lifetime risk of dementia, and five-fold increased risk of mortality from neurological disorders [30].

In the current study, we further elucidated the novel anti-inflammatory mechanism of GPER in OVX rats during the early phase of GCI as follows: (1) GPER activation induced by G1 suppresses the pro-inflammatory M1 phenotype and elevates the anti-inflammatory M2 phenotype of microglia after 1 and 3 days of reperfusion; (2) endogenous neuroprotectant BDE2 induced by GCI has anti-inflammatory and neuroprotective effects, at least partially depending on GPER; (3) the synergistic action of G1 and BDE2 elevates protein levels of anti-inflammatory factors IL-4, PPARγ, and BDNF during early reperfusion after GCI.

It is well known that GCI causes delayed neuronal death in the CA1 region of the hippocampus, which usually occurs after 3 to 7 days of reperfusion after GCI, and is accompanied by enhanced gliosis and overexpression of cytokines [31,32]. Therefore, pharmacological strategies to inhibit the pro-inflammatory M1 phenotype and/or promote the anti-inflammatory M2 phenotype of microglial cells prior to delayed neuronal death could reduce neuronal damage and improve neurological outcomes. Increasing evidence from our studies demonstrates that GPER activation with E2 or G1 exerts rapid neuroprotective effects in response to cerebral ischemia [26,33]. Moreover, a low dose of G1 (0.2 µg intracerebroventricularly) following middle cerebral artery occlusion (MCAO) provided significant neuroprotection in OVX mice [34]. Subsequent in vitro and in vivo studies documented that MCAO could upregulate the GPER level in microglia, and that a low dose of G1 protected ischemic neurons by inhibiting TLR4-mediated inflammation [23]. Additionally, a neuroprotective role of GPER activation induced by G1 has also been suggested in other chronic neurodegenerative disorders, such as Alzheimer’s disease [35] and multiple sclerosis [36]. In agreement with these reports, our current results show that GPER activation induced by G1 could potentially suppress early-phase inflammation by regulating the protein levels of M1/M2 microglial markers in OVX female rats following GCI. Specifically, we found that after 1 and 3 days of reperfusion after GCI, G1 administration significantly suppressed M1 markers such as MHCII, CD68, iNOS, and Iba1; robustly enhanced the protein expression of M2 markers CD206, Arginase 1, and IL1RA; and enhanced the protein levels of pro-survival PPARγ and BDNF. The results suggest that G1 acts as an anti-inflammatory/neuroprotective agent and might have clinical efficacy in cerebral ischemia. The mechanisms underlying the G1 regulation of the protein levels of these anti-inflammatory and neuroprotective factors was not examined in our current study. The G1 regulation of the factors could be due to either the regulation of gene transcription, and/or the modulation of the translation or degradation of the factors. Interestingly, previous work by our group showed that following GCI, G1 can increase the activation of extracellular signal kinase (ERK) and phosphoinositide 3 kinase (PI3K) in the hippocampus [37], which can regulate the transcription and translation of many genes and proteins [38,39,40,41]. Thus, it is possible that the G1 regulation of the anti-inflammatory and neuroprotective factors in the present study could be due to its ability to regulate ERK and PI3K signaling to provoke changes in the transcription and/or translation of the factors. Further studies are needed to address this interesting possibility.

Noxious stimuli such as cerebral ischemia and traumatic brain injury trigger endogenous protective mechanisms to protect the brain. Increasing evidence demonstrates that the biosynthetic enzyme Aro is crucial for endogenous neuroprotection in females, producing estrogen from precursor androgens in the brain [42]. BDE2 has more recently emerged as a potent neuroprotectant that exerts extensive neuroprotective effects, including decreasing neurodegeneration, neuroinflammation, and apoptosis [42,43,44]. It was reported that brain injury results in the enhancement in Aro expression in reactive astroglia and a consequent increase in local estradiol production at injury sites in the brain [42,45,46]. In the MCAO mouse model, Aro-knockout female mice had more outstanding ischemic brain damage than OVX or intact wild-type mice after 22 h of reperfusion after MCAO [47]. A more recent study by Brann’s laboratory generated a glial fibrillary acidic protein promoter-driven aromatase-knockout (GFAP-ARO-KO) mouse model to deplete astrocyte-derived E2 in the brain, and the animals showed elevated neuronal damage, microglial activation, and cognitive impairment following GCI [18]. Our current study adds to these important findings by further confirming that after 1 day of reperfusion after GCI, the protein expression and activity of Aro are upregulated, and the BDE2 level is significantly increased. Furthermore, our results suggest that GPER helps to mediate the neuroprotective and anti-inflammatory effects of Aro-BDE2 signaling, as treatment with G36, a GPER-specific antagonist, not only enhanced the fluorescence intensity of microglial marker Iba1 but also attenuated the number of NeuN-positive cells and the expression of anti-inflammatory factors IL4, PPARγ, and BDNF. Additionally, our findings suggest that GPER activation may involve BDE2, as treatment with an aromatase inhibitor markedly reversed G1-induced increases in M2 phenotype marker proteins IL4 and PPARγ, and the neuroprotective factor, BDNF. Of course, we cannot rule out the potential roles of ERα or/and -β in mediating BDE2 neuroprotection; after all, increasing data have proved that ERα/β have important roles against ischemic insults [48,49,50]. Thus, further work about the linkage between BDE2 and ERα/β in the GCI model, and even in intact (OVX) animals, needs to be pαformed in follow-up studies.

A proposed summary diagram depicting the interaction of GPER and Aro-BDE2 signaling in the regulation of anti-inflammatory actions after GCI is presented in Figure 7. As illustrated in Figure 7, our study provides evidence that BDE2 exerts a neuroprotective role, which at least in part depends on GPER, and that G1 acts synergistically with Aro-BDE2 signaling to attenuate early-stage inflammation by promoting the shift in the M1/M2 microglial markers in the hippocampal CA1 region following GCI. Taken together, the study suggests that G1 might be an effective agent to prevent early inflammation, which effectively protects hippocampal neurons against secondary delayed neuronal death following GCI.

## 5. Conclusions

Our study demonstrates that GPER activation can suppress early inflammation induced by GCI by attenuating pro-inflammatory factors while enhancing the expression of anti-inflammatory factors in the hippocampus. The beneficial effect appears to involve BDE2 mediation, as it is abolished by an aromatase inhibitor. Conversely, BDE2 neuroprotection appears to involve GPER signaling, as it is blocked by the GPER antagonist, G36.

## Figures and Tables

**Figure 1 biology-12-00099-f001:**
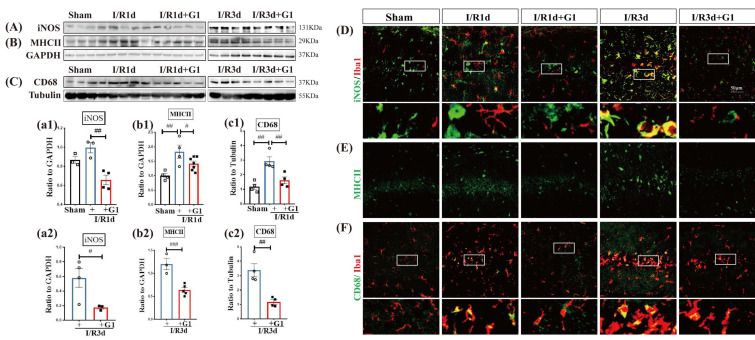
GPER activation induced by G1 suppressed M1 phenotypic markers of microglia in the ovariectomized rat hippocampal CA1 region following GCI. Hippocampal CA1 protein samples from animals in the sham group and animals 1 day and 3 days after GCI treated with or without G1 were subjected to Western blot analysis of microglial M1 markers iNOS (**A**,**a1**,**a2**), MHCII (**B**,**b1**,**b2**), and CD68 (**C**,**c1**,**c2**). (**D**–**F**) Brain coronal sections were used to perform immunofluorescence staining. Representative photomicrographs of (**D**) iNOS (green) and Iba1 (red), (**E**) MHCII (green), and (**F**) CD68 (green) and Iba1 (red) staining are shown. Data are shown as the means ± SEs from three to seven independent samples per group (*N* = 3–6). ^#^
*p* < 0.05, ^##^
*p* < 0.01 and ^###^
*p* < 0.001, Scale bar, 50 μm. Magnification, 40×. iNOS, inducible nitric oxide synthase; Iba1, Ionized calcium binding adaptor molecule 1; I/R, ischemia reperfusion. White Box in subfigure D, F, GPER, G-protein-coupled estrogen receptor 1.

**Figure 2 biology-12-00099-f002:**
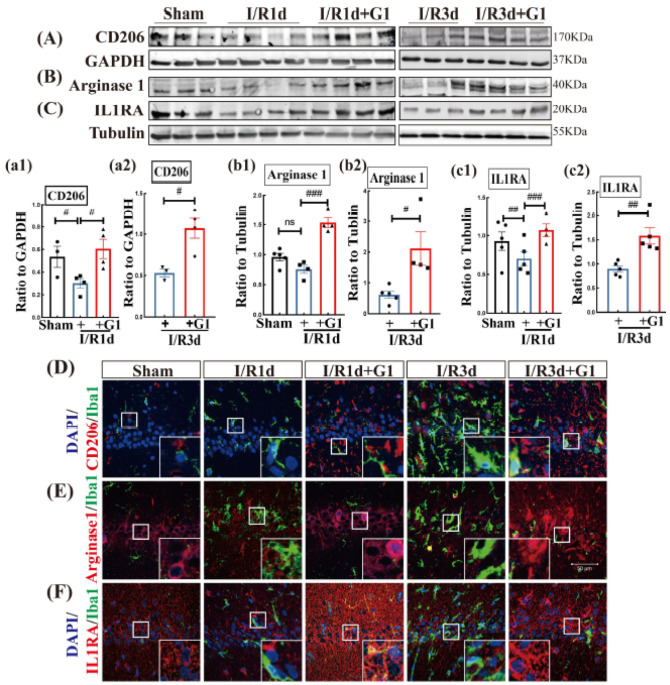
G1 administration enhanced M2 phenotypic marker proteins of microglia in the ovariectomized rat hippocampal CA1 region following GCI. Western blot analysis results of M2 markers CD206 (**A**,**a1**,**a2**), Arginase 1 (**B**,**b1**,**b2**), and IL1RA (**C**,**c1**,**c2**) are shown. Coronal brain sections were subjected to double immunofluorescence staining of Iba1 (green) and (**D**) CD206 (red), (**E**) Arginase 1 (red), and (**F**) IL1RA (red). DAPI (blue) was used to stain the nuclei. Data are shown as the means ± SEs from three to five independent samples per group (*n* = 3–5). ^#^
*p* < 0.05, ^##^
*p* < 0.01, and ^###^
*p* < 0.001. ns, no significance. Scale bar, 50 μm. Magnification, 40×. IL1RA, Interelukin-1 receptor antagonist; Iba1, Ionized calcium binding adaptor molecule 1. White Box in subfigure D–F, GPER, G-protein-coupled estrogen receptor 1.

**Figure 3 biology-12-00099-f003:**
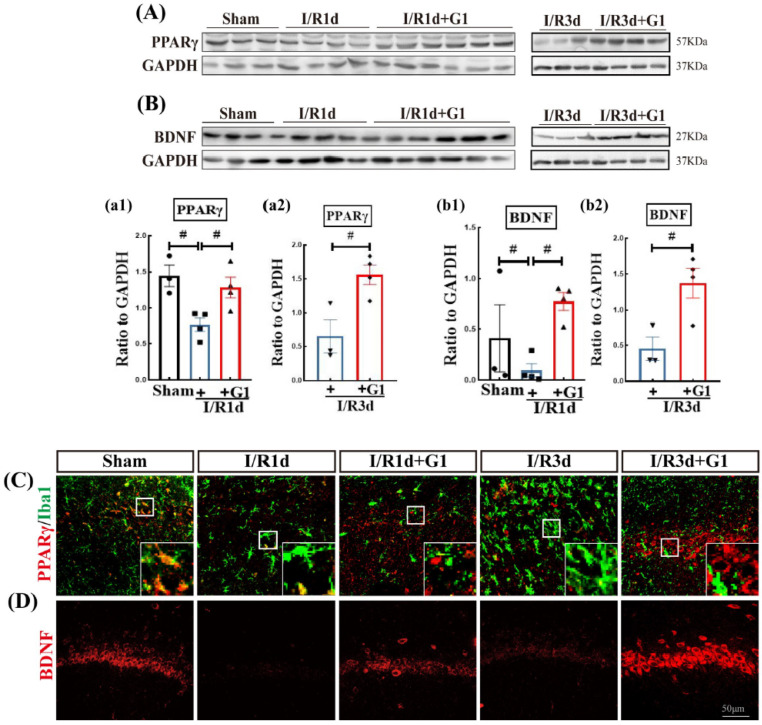
G1 increased the levels of BDNF and PPARγ protein in the ovariectomized rat hippocampal CA1 region after GCI. Western blot analysis results of PPARγ (**A**,**a1**,**a2**) and BDNF (**B**,**b1**,**b2**) are shown. Coronal brain sections were subjected to immunofluorescence staining for PPARγ (red) and Iba1 (green) (**C**) and for BDNF (red) (**D**). Data are shown as the means ± SEs from three to five independent samples per group (*n* = 3–5). ^#^
*p* < 0.05. Scale bar, 50 µm. Magnification, 40×. BDNF, Brain-derived neurotrophic factor; Iba1, Ionized calcium binding adaptor molecule 1; PPARγ, Peroxidase Proliferator-Activated Receptor-γ. White Box in subfigure C, GPER, G-protein-coupled estrogen receptor 1.

**Figure 4 biology-12-00099-f004:**
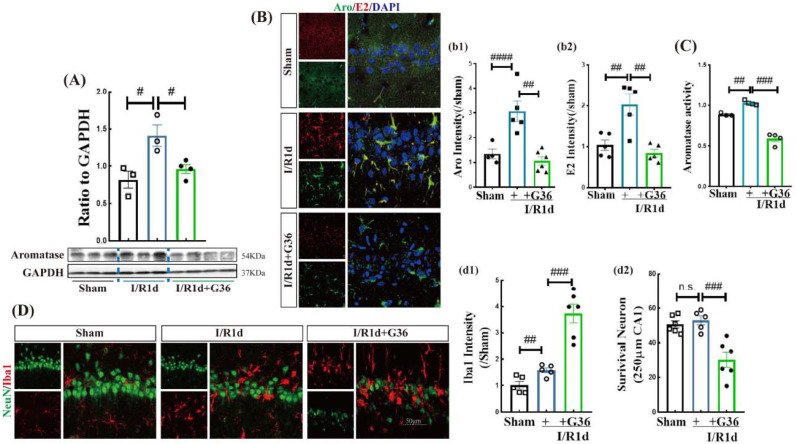
GPER was required for the neuroprotective effect of Aromatase-BDE2 signaling in the ovariectomized rat hippocampal CA1 region after GCI. (**A**) Western blot analysis was used to detect changes in aromatase in the sham, one day of reperfusion (I/R1d), and I/R1d + G36 treatment groups. (**C**) Effect of G36 on Aro activity, as determined using an enzyme-linked immunosorbent assay (ELISA). Representative photographs of double immunofluorescence staining of (**B**) Aro (green) and E2 (red), and (**D**) NeuN (green) and Iba1 (red) in the indicated groups are shown. DAPI (blue) was used to stain the nuclei. A quantitative analysis of the fluorescence intensity ratio of Aro (**b1**), E2 (**b2**), and Iba1 (**d1**) to the sham group is shown. (**d2**) Quantification was performed by counting the number of NeuN-positive neurons per 250 µm segment in the medial CA1 pyramidal cell layer. (*n* = 3–6 per group). ^#^
*p* < 0.05, ^##^
*p* < 0.01, and ^###^
*p* < 0.001. and ^####^
*p* < 0.0001, n.s., no significance. Scale bar, 50 μm. Magnification, 40×. GPER, G-protein-coupled estrogen receptor 1; BDE2, brain-derived estrogen; Aro, Aromatase; E2, 17β-estradiol; Iba1, Ionized calcium binding adaptor molecule 1.

**Figure 5 biology-12-00099-f005:**
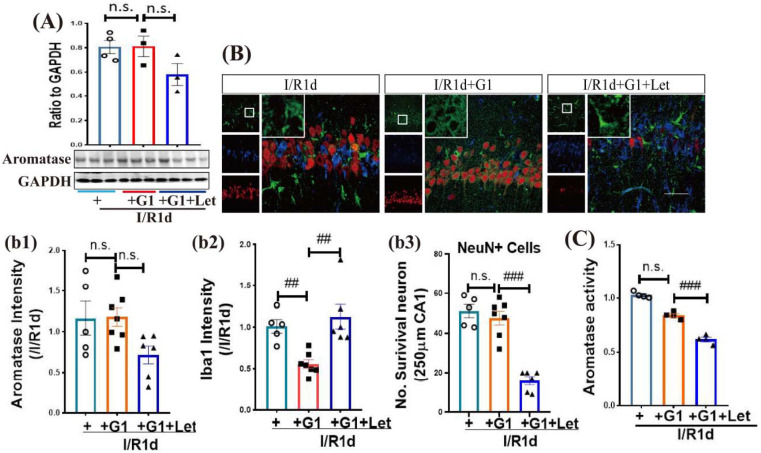
The anti-inflammatory effect of G1 after global cerebral ischemia was dependent on BDE2 in the ovariectomized rat hippocampal CA1 region. (**A**) Hippocampal CA1 samples from the one day of reperfusion (I/R1d), I/R1d+G1, and G1+Let groups were subjected to Western blot analysis of Aro. Quantitative data are expressed as the ratios to GAPDH. (**B**) Representative photographs of triple immunofluorescence staining of Aro (green), Iba1 (blue), and NeuN (red). Quantitative analysis results of fluorescence intensity of Aro (**b1**) and Iba1 (**b2**) staining in the hippocampal CA1 region, expressed in arbitrary units as ratios to the I/R1d group, were obtained. (**b3**) The number of surviving neurons was quantified by counting the number of NeuN-positive neurons per 250 µm segment in the medial CA1 pyramidal cell layer. (**C**) The enzyme-linked immunosorbent assay analysis showed the effect of Let on aromatase activity with or without G1 treatment. Data are represented as the means ± SEs from five to seven independent samples (*n* = 3–5 in each group). ^##^
*p* < 0.01, and ^###^
*p* < 0.001. n.s. no significance. Scale bar, 50 μm. Magnification, 40×. Aro, aromatase; Let, letrozole; I/R, ischemia reperfusion; ns, not significant; Iba1, Ionized calcium binding adaptor molecule 1. White Box in subfigure B, GPER, G-protein-coupled estrogen receptor 1.

**Figure 6 biology-12-00099-f006:**
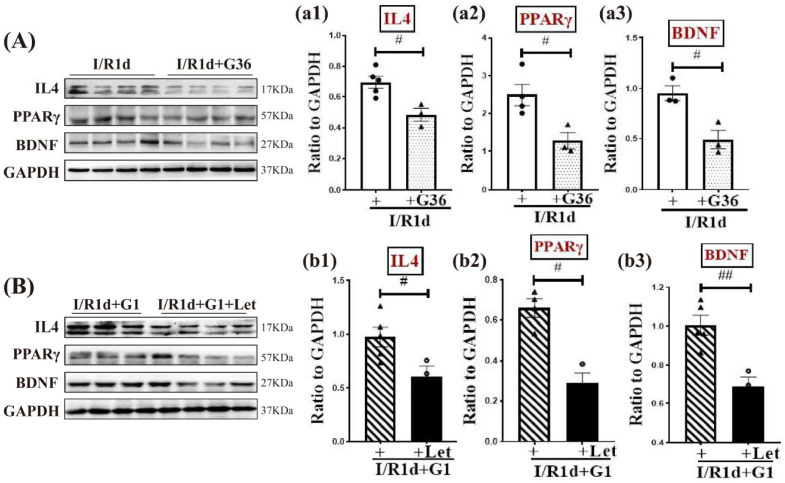
Synergistic action of G1 and BDE2 promoted microglial polarization to the M2 phenotype after 1 day of reperfusion in the ovariectomized rat hippocampal CA1 region. (**A**,**B**) Representative bands of Western blot analysis of IL4, PPARγ, and BDNF in the indicated groups. GAPDH was used as the loading control. Quantification of blots is depicted in (**a1**–**a3**) and (**b1**–**b3**). Data are expressed as ratios to GAPDH (means ± SEs, *n* = 3–5 in each group). ^#^
*p* < 0.05 and ^##^
*p* < 0.01. BDE2, brain-derived estrogen; IL, Interleukin; BDNF, Brain-derived neurotrophic factor.

**Figure 7 biology-12-00099-f007:**
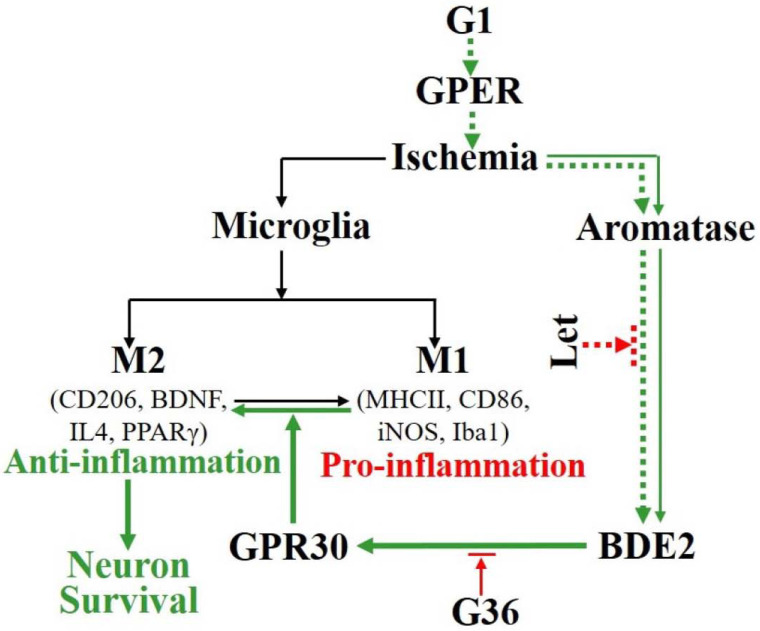
Summary diagram of the proposed mechanism underlying the synergistic regulation of microglial polarization induced by G1 and BDE2 that attenuates neuronal impairment in the ovariectomized rat hippocampal CA1 region following GCI. Cerebral ischemia induces Aro-BDE2, which promotes protein expression of M2 markers via GPER, as evidenced by the inhibition caused by GPER antagonist G36. GPER activation induced by G1 reverses microglial polarization from the pro-inflammatory M1 to the anti-inflammatory M2 phenotype after 1 and 3 days of reperfusion after GCI. Meanwhile, Aro-BDE2 signaling induced by G1 is blocked by Let and decreases the protein expression of M2 phenotype markers. Together, both G36 and Let treatments decrease neuronal survival in the hippocampal CA1 region in the early stage of GCI. Aro, aromatase; GCI, global cerebral ischemia; BDE2, brain-derived estrogen; GPER, G-protein-coupled estrogen receptor 1; Let, letrozole.

## Data Availability

Please contact corresponding author at Ruimin-wang@163.com if necessary.

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
