# Peer review of "Anti-Inflammatory Actions of G-Protein-Coupled Estrogen Receptor 1 (GPER) and Brain-Derived Estrogen Following Cerebral Ischemia in Ovariectomized Rats"

_biology, 2023, doi:10.3390/biology12010099_

Round 1

Reviewer 1 Report

This manuscript from Xu et al. demonstrates that the novel G-protein coupled estrogen receptor (GPER) agonist, G1, suppresses inflammation and reduces neuronal loss after experimental global cerebral ischemia, at least in part, via regulation of brain-derived estradiol.  Overall, the work is timely and supports the growing concept that estrogens produced locally may exert a neuroprotective role after acute brain injury.  The paper is well written and clearly presented.  While some studies may be considered somewhat  incremental and observational in nature, the work overall is important and will continue to advance the field.  

Despite these clear strengths, a few minor points for the authors to consider:

1. All drugs were utilized at a single dose and therapeutic window.  Could the authors clarify why these doses were selected?  Pharmacological agents often exhibit off target effects and other drug interactions that could be considered.  This caveat could be presented, as well. Finally, is it known whether the drugs cross the blood-brain barrier?

2.  FIg 1a-c.  It was difficult to see how the treatment groups were labeled in the Western blot figures.  It appears a faint line may be present to show which group each band belongs, but this should be more clearly presented.

3. A distinct limitation of the work is the absence of any functional and/or long-term outcome measures.  No behavioral studies were included and longer term time points beyond d3 were absent.  Can the authors comment on whether these observed changes reflect any functional deficits?  Similarly, do these effects persist beyond the acute time points - is it possible that BDE simply delays, but does not actually reverse, neuronal loss?   In addition, M1/M2 are not simply good vs bad...rather M1 microglia may have beneficial effects at certain timepoints, but maybe not in the long-term.  While these are recognized field-wide limitations, some discussion may be warranted.

Author Response

Many thanks for your kind comments and excellent suggestions. Please find the uploaded documents about our responses to your comments and the revised manuscript. 

Reviewer 2 Report

This paper studied the anti-inflammatory effects of E2 in hippocampal CA1.

In order to improve the completeness of the paper, I would like to suggest some amendments.

Lines 61-63: Reference required.

Note the abbreviation in the abstract: 3d needs correction

Line 83: full term required for TLR 4

Line 534: Change x to upper case.

Line 189, 224, 250, 273, 307, 375 Correct the p value expression to lower case.

Please, write the full term of NeuN.

- OVX is an estrogen deficiency animal model. Can this study be applied to premenopausal cases with abundant production of estrogen, especially E2? If the subject of the study cannot be generalized, it is reasonable to narrow the research subject(title) to postmenopausal women.

-In your study design design, you have combined a GPER agonist, a GPER antagonist, and an aromatase inhibitor. A supplementary explanation is needed to easily convey to the reader what phenomena or lesions in the human body this combination is designed to simulate

-(GPER/GPR30) is an estrogen receptor that induces a rapid non-genomic response. It shows a completely different response from the ER alpha and ER beta receptors that induce classical genomic responses. Genomic action requires 6-12 hours for expression because gene transcription-translation is involved, but non-genomic action takes place within seconds to minutes after estrogen response. It is necessary to logically highlight the meaning and importance of non-genomic reactions that occur in a short time in this study. Please explain that the design of this study is valid considering the non-genomic response.

- In the OVX model, supplement the comparative analysis of total serum estradiol level and the amount of estradiol produced in the hippocampal CA1 region of ovariectomized rats and aromatase activity, as well as comparative data on GPER, ER alpha, ER beta, and binding affinity. Please supplement the explanation of how much estradiol is produced and reacts with GPER.

-In the Material and Method section, the description of the measurement of aromatase activity is lacking. Supplement the overall information such as the machine used, the sample used in the experiment, and the specimen.

Author Response

Thanks for your kind comments and excellent suggestions on our article. Please find uploaded documents about our response to your comments and revised manuscript. 

Reviewer 3 Report

Title: Synergism Between G-Protein-Coupled Estrogen Receptor 1 2 (GPER) Signaling and Brain-Derived Estrogen in Exerting Anti- 3 Inflammatory Actions in the Hippocampus Following Cerebral Ischemia

In this study, Xu et al demonstrate the role of G1, an agonist of the GPER receptor, in protecting against inflammation and apoptosis, and make a proposal for a microglial regulatory mechanism mediated by G1 and BDE2.

I believe the work is well written, with interesting results that could be accepted with minor changes.

ü  The title of the article appears to be lengthy and ineffective in attracting the attention of those who are interested in the subject.

ü  The highlights before the abstract are interesting, but they should be part of the conclusions that the article lacks due to the density of the research work.

ü   The introduction is current and adequate for discussing the subject.

ü  The materials and methods are presented in all the detail required to comprehend the results, but I believe it is important to mention the ethical considerations (committee endorsement to carry out experimentation with animals).

ü  The results are presented in an organized manner and are convincing since the methodologies used, Western blot analysis and immunofluorescence staining, are complementary.

ü  The discussion appears to integrate the obtained results and highlights the proposed mechanism.

ü  As previously stated, general conclusions are required in which the major contributions to the knowledge of the developed research work are highlighted.

Author Response

Thanks for your positive comments on our article. Please find the uploaded documents about our response to your review and revised manuscript.

Round 2

Reviewer 2 Report

Question) OVX is an estrogen deficiency animal model. Can this study be applied to premenopausal cases with abundant production of estrogen, especially E2? If the subject of the study cannot be generalized, it is reasonable to narrow the research subject (title) to postmenopausal women.

My opinion) In postmenopausal women, estrogen is no longer an endocrine factor but is produced in several extragonadal sites and acts locally in these sites in a paracrine and endocrine manner. Comparison of the response when E2 is abundantly produced in the Ovary before menopause and the response when it is produced in the extragonadal site after menopause, analysis of the mechanism Supplementation is necessary, but no modifications have been made. And since this study was conducted only on the E2-deficient OVX animal model, the animal model should be clearly specified in the title, and care should be taken to let readers know at once whether it is a human study or an animal study. Since it has not been modified at all, publication of this paper is rejected.

Question) -(GPER/GPR30) is an estrogen receptor that induces a rapid non-genomic response. It shows a completely different response from the ER alpha and ER beta receptors that induce classical genomic responses. Genomic action requires 6-12 hours for expression because gene transcription-translation is involved, but non-genomic action takes place within seconds to minutes after estrogen response. It is necessary to logically highlight the meaning and importance of non-genomic reactions that occur in a short time in this study. Please explain that the design of this study is valid considering the non-genomic response.

My opinion) Looking at the answers to this question, authors do not present data or develop logic about their research. The logic of a researcher's own research cannot be replaced by other people's experimental results and papers. There is no clear explanation and data that the design of the study and the time and process of the experiment are valid for the non-genomic response.

I object to the publication of this paper.

Question) - In the OVX model, supplement the comparative analysis of total serum estradiol level and the amount of estradiol produced in the hippocampal CA1 region of ovariectomized rats and aromatase activity, as well as comparative data on GPER, ER alpha, ER beta, and binding affinity. Please supplement the explanation of how much estradiol is produced and reacts with GPER.

My opinion) In this experiment, the level of circulating E2 and the level of E2 produced in the brain are not explained. E2 naturally decreased in the OVX model, and production of E2 in non-gonadal regions would increase. In the study of global ischemia, the OVX model is not a receptor knockout model. Therefore, GPER, ER alpha, and ER beta all coexist. All of these receptors are expressed in the brain. The authors did not present to the readers evidence that GPER is dominant. In the global ischemia-induced cell death model, the important roles of ER alpha and ER beta have already been identified (Ref). As there is no supplement to the main text, publication of this paper is rejected.

References)

Miller NR, Jover T, Cohen HW, Zukin RS, Etgen AM: Estrogen can act via estrogen receptor alpha and beta to protect hippocampal neurons against global ischemia-induced cell deathEndocrinology 2005; 146: 3070–3079.

Dai X, Chen L, and Sokabe M: Neurosteroid estradiol rescues ischemia-induced deficit in the long-term potentiation of rat hippocampal CA1 neurons. Neuropharmacology 2007; 52: 1124–1138.

Zhang QG, Han D, Wang RM, Dong Y, Yang F, Vadlamudi RK, Brann DW: C terminus of Hsc70-interacting protein (CHIP)-mediated degradation of hippocampal estrogen receptor-alpha and the critical period hypothesis of estrogen neuroprotection. Proc Natl Acad Sci U S A 2011; 108: E617–E624. 

Author Response

Dear Reviewer,

We thank you for your very helpful and thoughtful comments/suggestions on our paper. Please see the uploaded document which is our responses to the questions. Thanks!
